# Impact of Cross-Tie Material Nonlinearity on the Dynamic Behavior of Shallow Flexible Cable Networks

## Amir Younespour * and Shaohong Cheng

Department of Civil and Environmental Engineering, University of Windsor, Windsor, ON N9B 3P4, Canada; shaohong@uwindsor.ca
* Correspondence: younespa@uwindsor.ca

**Abstract:** Cross-ties have proven their efficacy in mitigating vibrations in bridge stay cables. Several factors, such as cross-tie malfunctions due to slackening or snapping, as well as the utilization of high-energy dissipative materials, can introduce nonlinear restoring forces in the cross-ties. While previous studies have investigated the influence of the former on cable network dynamics, the evaluation of the impact of nonlinear cross-tie materials remains unexplored. In this current research, an existing analytical model of a two-shallow-flexible-cable network has been extended to incorporate the cross-tie material nonlinearity in the formulation. The harmonic balance method (HBM) is employed to determine the equivalent linear stiffness of the cross-ties. The dynamic response of a cable network containing nonlinear cross-ties is approximated by comparing it to an equivalent linear system. Additionally, the study delves into the effects of the cable vibration amplitude, cross-tie material properties, installation location, and the length ratio between constituent cables on both the fundamental frequency of the cable network and the equivalent linear stiffness of the cross-ties. The findings reveal that the presence of cross-tie nonlinearity significantly influences the in-plane modal response of the cable network. Not only the frequencies of all the modes are reduced, but the formation of local modes is delayed to a high order. In contrast to an earlier finding based on a linear cross-tie assumption, with nonlinearity present, moving a cross-tie towards the mid-span of a cable would not enhance the in-plane stiffness of the network. Moreover, the impact of the length ratio on the network in-plane stiffness and frequency is contingent on its combined effect on the cross-tie axial stiffness and the lateral stiffness of neighboring cables.

**Keywords:** cable; nonlinear cross-tie; material nonlinearity; cable network; dynamic behavior



## 1. Introduction

The continuous growth in the span length of cable-stayed bridges leads to longer stay cables, which are more flexible and prone to large-amplitude vibrations [1–4]. Besides the aerodynamic treatment of cable surface [5,6] and installing supplemental damping devices [7–12], the cross-tie solution is another typical countermeasure for suppressing unfavorable vibrations of bridge stay cables [13–16]. Cross-tie(s) can be employed to connect a vulnerable cable transversely to its neighboring ones and form a cable network, as shown in Figure 1. Studies in recent years show that such a measure would enhance the in-plane stiffness of the vulnerable cable [17,18], introduce additional damping to the connected cables [19,20], increase the modal mass of the vibrating system [21], and redistribute energy contained in the consisting cables [22]. In conjunction with these, it has been confirmed experimentally that the cross-tie solution could also improve the aerodynamic stability of a cable by eliminating the wake effects induced by the proximity of neighboring cables [23,24]. On the other hand, however, the cross-tie solution has a few critical drawbacks, including limitations in suppressing out-of-plane cable vibrations and the inability to directly dissipate energy from oscillating cables unless the cross-tie material itself can dissipate energy [25,26].

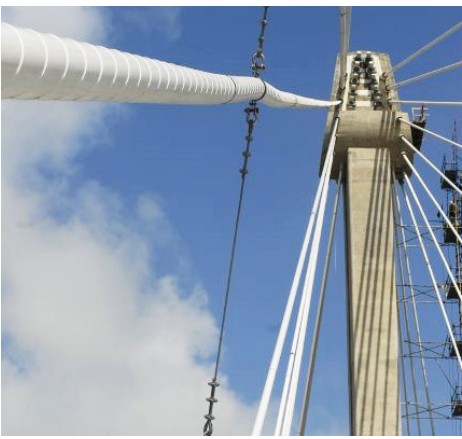

**Figure 1.** Cable cross-tie [27].

In the majority of the existing cable network analytical models, the cross-tie is assumed to be either a rigid or a linear elastic connector (e.g., [14,16,26,28]). However, their applications on real cable-stayed bridges show that cross-ties typically exhibit nonlinear behavior [3,25], and a slight change in their properties could have a considerable impact on the dynamic response of cable networks. Under large-amplitude cable vibrations, the cross-tie could fail due to different mechanisms. For example, if two adjacent stays in a cable network vibrate out-of-phase, it may cause either slack or snap of the cross-tie, which could lead to a reduction in the local stiffness of the cross-tie. Due to this fact, the restoring force in the cross-tie cannot be properly transferred to the main cables [29]. It is worth noting that the relationship between the cross-tie restoring force and the cross-tie axial deformation is nonlinear when slackening and snapping occurs. In addition, high energy dissipative material, such as rubber [30], can be used as cross-tie material, which, in general, exhibits nonlinear behavior. The presence of these potential sources of cross-tie nonlinearity could engender non-negligible discrepancies between the actual dynamic response of a cable network and that predicted based on the linear cross-tie assumption. Thus, it is necessary to take into account the effect of cross-tie nonlinearity when formulating the cable network analytical model for investigating its dynamic behavior.

A number of studies investigated the performance of cable network systems when the cross-tie nonlinear behavior is considered and when high-energy dissipative material is used as cross-tie material. In an earlier experimental study by Yamaguchi and Nagahawatta [20], the damping effect of cross-tie and connected cables was evaluated by taking into account the nonlinear interaction between the cross-tie and the cables. Results showed that the impact of cross-tie damping on the effectiveness of cable vibration control can be significant if more flexible and energy-dissipative cross-ties were employed. In addition, the nonlinear impact due to the interaction between the secondary cable and main cables on the cable network response was studied by a few researchers through experimental tests [15,31]. In addition, by adopting highly energy-dissipative material for the cross-tie, the impact of cross-tie stiffness on the in-plane vibration mitigation of cable network systems was explored in an experimental study [30]. Since this kind of energy-dissipative material has a nonlinear property, an energy-based method was employed to analytically evaluate the equivalent linear damping ratio of the cable network when the nonlinear cross-tie was used. However, a detailed analytical study to show how the dynamic response of a cable network would be affected by using an energy-dissipative cross-tie is not available. Later, Giaccu and Caracoglia [29] proposed a nonlinear spring element with cubic stiffness combined in parallel with a linear stiffness spring element to model the nonlinear restoring-force mechanism of the cross-tie. This nonlinear cross-tie model was applied to analytically study the dynamics of cable networks in the presence of nonlinear interactions between the cross-tie and the connected cables. Subsequently, a nonlinear spring element with a more general power-law form of stiffness was employed to account for the nonlinear restoring-force

in the cross-tie [32]. In these investigations, the original cable network with a nonlinear cross-tie was replaced by an equivalent linearized system using the equivalent linearization method (ELM). The modal frequencies of the original system were determined by solving the eigenvalue problem of the equivalent linearized system. It is known that the most common causes of cable vibrations are wind and rain-wind excitations. Due to the random nature of these excitation sources, the amplitude of the induced cable vibrations could be affected by various uncertainties. Therefore, stochastic analysis was performed [33,34] for cable networks to address these uncertainties. It has been highlighted that the impact of cross-tie nonlinearity on the performance of a network system can be significant, contingent upon factors, such as cable tension and cross-tie pre-tension. Moreover, Giaccu et al. [35] proposed a discrete mass model for cable networks. Their study encompassed an exploration of the dynamics of a three-cable network featuring nonlinear cross-ties, as well as the incorporation of considerations regarding initial pre-tension force and its subsequent loss. It was demonstrated that a significant reduction in the cross-tie stiffness might occur when the cross-tie pre-tension force was smaller than the tension force in the stay. In the studies by Giaccu et al. [29,32–35], the focus was on the nonlinear interaction between the cross-tie and main cables due to cross-tie malfunctions when slackening and/or snapping occurs, but the cross-tie nonlinearity caused by its material property was not considered. However, when energy-dissipative material is used for the cross-tie, the presence of material nonlinearity in the cross-tie and its impact on the cable network response cannot be ignored.

There are a number of nonlinear techniques, such as nonlinear normal mode and the perturbation method, which can properly take care of the problem related to the geometric nonlinearity in a single cable [36–38]. However, extending their applications to a complex cable network system is not warranted in a simple way due to the challenge in the implementation. The existing cable network analytical models with nonlinear cross-ties employed the ELM to examine the dynamics of cable network systems in the presence of a cross-tie malfunction caused by slackening and/or snapping [29,32]. Results showed that while the ELM was acceptable for cable network modeling and practical applications, its accuracy was inadequate. Therefore, to address the impact of the cross-tie material nonlinearity, a technique that is simple to implement, yet can still achieve a desired accuracy, is required. The harmonic balance method (HBM) is considered to be one of the effective methods that is capable of handling problems associated with strongly nonlinear systems [37,39]. This method was originally developed to study structural vibrations of elastic systems and used for the calculation of periodic solutions of nonlinear systems [40]. The HBM is based on the assumption that the time domain response of a system can be expressed in the form of a truncated Fourier series. The series coefficients can be determined by solving a set of equivalent linear equations. Another advantage of the HBM is that it can be combined with other methods, such as the Volterra series approach, to further improve its accuracy [41].

In this present investigation, the Harmonic Balance Method (HBM) is employed to develop an analytical model for a two-cable network, taking into account the material nonlinearity of the cross-tie. This involves refining an existing analytical model that pertains to a cable network consisting of two flexible shallow cables [42]. To incorporate HBM into the proposed analytical model, the non-dimensional force-displacement relationship that characterizes the cross-tie material property is adapted using a piecewise power series polynomial format. Two distinct types of material nonlinearity, in particular, strain-softening and strain-hardening behaviors, are taken into consideration. A comparative analysis will be conducted between the responses of cable networks featuring linear and nonlinear cross-ties. Furthermore, beyond evaluating the amplitude of cable vibrations, the study will delve into the impact of other system characteristics. These include the cross-tie material property, installation location, and the relative length ratio between the neighboring cable and the designated cable. These factors will be examined with regard to

their influence on both the equivalent linear stiffness of the cross-tie and the fundamental frequency of the cable network.

## 2. Formulation of Analytical Model

Illustrated in Figure 2 is a configuration of a cable network comprising two horizontally positioned flexible shallow cables. These cables are linked by a transverse cross-tie. Main cable 1 is assumed to be the target cable, which is vulnerable to dynamic excitations and could experience large-amplitude oscillations. Main cable 2 serves as a colleague cable to assist in vibration control of the target cable. The $k^{th}$ cable ($k = 1$, 2) is characterized by its length $L_k$ ($L_1 \geq L_2$), the mass per unit length $m_k$, the chord tension $H_k$, and the bending stiffness $E_k I_k$, where $E_k$ and $I_k$ are the Young's modulus and the moment of inertia of the cable cross-section, respectively. The sag at the mid-span of the $k^{th}$ cable is represented by $d_k$. Both cables are fixed at two ends. The horizontal displacement of cable 2 from the left and right side of cable 1 is represented as $O_L$ and $O_R$, respectively ($O_L \neq O_R$). A transverse cross-tie is installed at a distance $l_{1,1}$ from the left end of the target cable ($l_{1,1} \leq l_{1,2}$). It divides each cable into two segments. In addition, it is assumed that vibrations of the main cables are dominated by the in-plane transverse motion, whereas that of the cross-tie is dominated by the longitudinal oscillation. Thus, this study disregards the longitudinal motion of the main cables and the transverse motion of the cross-tie. Notably, a positive value is ascribed to the downward vertical displacement.

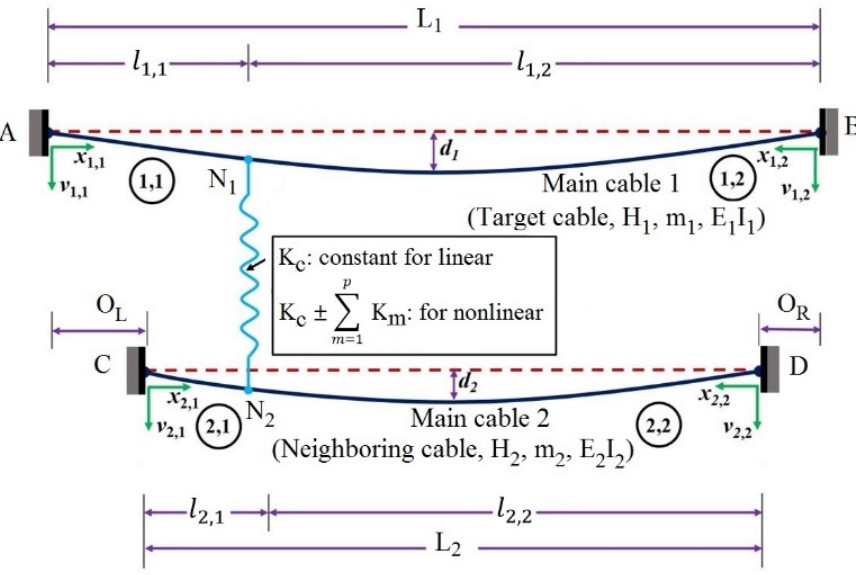

**Figure 2.** Schematic diagram of the mathematical model of a two-shallow-flexible-cable network with a nonlinear cross-tie.

The analytical model of a flexible-shallow-cable network with the same layout as that in Figure 2 has been derived by Younespour and Cheng [42] under the condition of a linear elastic cross-tie. However, for real cable networks on site, the imperfect transfer mechanism of the restoring force at the connection of the cross-tie and the main cable and the adoption of energy-dissipative material for the cross-tie would inevitably induce nonlinearity in the cross-tie behavior. To explore the influence of cross-tie material nonlinearity on the in-plane dynamic response of cable networks, in the current study, the analytical model developed in [42] will be extended to take into account the cross-tie material nonlinearity in the formulation. This would allow for a prediction of the dynamic response of cable network systems more realistically and accurately.

In the model formulation, the mass and the pre-tension of the cross-tie are not considered. If we assume that $\delta_c = [v_{1,1}(x,t) - v_{2,1}(x,t)]$ is the axial deformation of the cross-tie, $v_{k,j}(x,t)$ is the in-plane transverse displacement of the $j^{th}$ segment in the $k^{th}$ cable ($k = 1$, 2;

$j = 1, 2$), $K_c$ is the cross-tie linear axial stiffness, $F_u$ is the maximum force resisted by the cross-tie, and $\delta_0$ and $\delta_u$ are the total elastic deformation and the ultimate deformation of the cross-tie, respectively, the non-dimensional cross-tie restoring force $\overline{F}_c$ can be defined as follows:

$$\overline{F}_c(\overline{\delta}_c) = \begin{cases} \overline{K}_c \overline{\delta}_c \quad , & 0 < \overline{\delta}_c \leq \delta_0/\delta_u \\ \displaystyle\sum_{m=1}^{p} \overline{K}_m \overline{\delta}_c^{\,m} \quad , & \delta_0/\delta_u < \overline{\delta}_c \leq 1 \end{cases} \tag{1}$$

where $\overline{\delta}_c = \delta_c/\delta_u$ and $\overline{K}_c = K_c \delta_u/F_u$ are the non-dimensional cross-tie deformation and the non-dimensional cross-tie linear stiffness, respectively. $\overline{K}_m$ is the coefficient of the $m^{\text{th}}$ polynomial term, and $p$ is an integer representing the highest order of the polynomial term, which can be either odd or even and is determined based on the required accuracy of the approximated force-displacement relation of cross-tie material. It is worth mentioning that in the current study, the constitutive relationship of a type of nonlinear material is obtained by applying curve fitting to a set of force-displacement data of the material. Therefore, the proposed model is able to describe the elastic perfectly plastic behavior of a cross-tie as well.

The step-by-step time integration method is commonly used in analyzing strongly nonlinear vibration problems [43]. However, obtaining the frequency responses from this approach can be challenging. The Harmonic Balance Method (HBM) is based on the assumption that the dynamic response of a nonlinear system in the time domain can be expressed in the form of a truncated Fourier series. Unlike other nonlinear approaches, such as the perturbation method, HBM can offer satisfactory results even for strongly nonlinear systems and is applicable to complex nonlinear systems [37,39]. In addition, the accuracy of the HBM can be improved with a simple combination of other approaches if needed, such as the Volterra series approach [41]. In this study, the non-dimensional nonlinear restoring stiffness of the cross-tie in Equation (1) can be transformed into a non-dimensional equivalent linear stiffness using the HBM. This approach has been applied to various types of nonlinear problems as a means of approximating frequency response functions of nonlinear systems [44,45], of which the nonlinear quantity can be expressed as a truncated Fourier series based on the applied force and the resulting system response. In the current study, the in-plane transverse displacement of the $j^{\text{th}}$ segment in the $k^{\text{th}}$ cable can be expressed as [14,29]

$$v_{k,j}\left(x_{k,j}, t\right) = A_{Ek,j}\sin\left(\Omega\eta_k \frac{x_{k,j}}{L_k}\right)\sin(\omega t) \tag{2}$$

where $A_{Ek,j}$ is the approximated amplitude of the eigenfunction of the $j^{\text{th}}$ segment in the $k^{\text{th}}$ cable, $\eta_k = f_1/f_k$ is the frequency ratio between the target cable (cable 1) and cable $k$ ($k = 1, 2$), $\Omega = \pi f/f_1$ is the dimensionless system frequency of the studied cable network, $f_k$ is the fundamental frequency of main cable $k$ ($k = 1, 2$), $f$ is the fundamental frequency of the cable network, which needs to be determined, $\omega$ is the circular frequency of vibration, and $L_k$ is the length of the $k^{\text{th}}$ cable. The non-dimensional axial deformation of the cross-tie is expressed as $\overline{\delta}_c = \frac{1}{\delta_u}[v_{1,1}(l_{1,1}, t) - v_{2,1}(l_{2,1}, t)] = \frac{1}{\delta_u}Q\sin(\omega t)$, where $Q = A_{E1,1}\sin(\Omega\eta_1\varepsilon_{1,1}) - A_{E2,1}\sin(\Omega\eta_2\varepsilon_{2,1})$. $\varepsilon_{k,j} = l_{k,j}/L_k$ is the segment ratio, and $l_{k,j}$ is the length of the $j^{\text{th}}$ segment in the $k^{\text{th}}$ cable, and the non-dimensional cross-tie nonlinear restoring force can be obtained as

$$\overline{F}_c(\overline{\delta}_c) = \overline{K}_{eq}\overline{\delta}_c = \frac{\overline{K}_{eq}}{\delta_u}Q\sin(\omega t) \tag{3}$$

where $\overline{K}_{eq}$ is a non-dimensional equivalent linear axial stiffness of the nonlinear cross-tie, which can be determined using the HBM. To apply the HBM, $\overline{F}_c(\overline{\delta}_c)$ is expressed as a truncated Fourier series [46], i.e.,

$$\frac{\overline{K}_{eq}}{\delta_u}Q\sin(\omega t) = a_0 + \sum_{n=1}^{q} a_n \cos(n\omega t) + \sum_{n=1}^{q} b_n \sin(n\omega t) \tag{4}$$

where $q$ is an integer that is determined by the required accuracy of the approximation, and $a_0 = \frac{1}{2\pi}\int_0^{2\pi} \overline{F}_c(\overline{\delta}_c)d(\omega t)$, $a_n = \frac{1}{\pi}\int_0^{2\pi} \overline{F}_c(\overline{\delta}_c)\cos(n\omega t)d(\omega t)$, and $b_n = \frac{1}{\pi}\int_0^{2\pi} \overline{F}_c(\overline{\delta}_c)\sin(n\omega t)$ $d(\omega t)$ $(n = 1, 2, \ldots, q)$ are the HBM solution Fourier coefficients [46].

The HBM finds frequent applications in cases where the amplitude $A_n = \sqrt{a_n^2 + b_n^2}$ of the higher-order harmonics ($n > 1$) and the constant term $a_0$ is substantially lesser than the amplitude $A_1$ of the harmonic term [41,47]. The truncated Fourier series must yield convergence to the anticipated value, aligning with the precision sought for the approximation; otherwise, more terms are needed. Since large value of $n$ will yield small integration values of $a_n$ and $b_n$, for the purpose of HBM, the most important terms of the expansion in Equation (4) are the fundamental terms (i.e., $a_0$, $a_1$, and $b_1$). In addition, noticing that all terms on the left-hand side of Equation (4) contain only the $\sin(\omega t)$ term, it yields $\frac{\overline{K}_{eq}}{\delta_u}Q = b_1$, where $b_1$ can be calculated over a period $[0, 2\pi]$ as

$$b_1 = \frac{1}{\pi}\int_0^{2\pi} \overline{F}_c(\overline{\delta}_c)\sin(\omega t)d(\omega t) \tag{5}$$

Substitute Equation (5) into $\frac{\overline{K}_{eq}}{\delta_u}Q = b_1$ and considering the cross-tie restoring force function $\overline{F}_c(\overline{\delta}_c)$ in Equation (1), it yields

$$\overline{K}_{eq} = \frac{\delta_u b_1}{Q} = \frac{\delta_u}{\pi Q}\left\{\underbrace{\int_0^{2\pi} \overline{F}_c(\overline{\delta}_c)\sin(\omega t)d(\omega t)}_{0<\overline{\delta}_c\leq\delta_0/\delta_u} + \underbrace{\int_0^{2\pi} \overline{F}_c(\overline{\delta}_c)\sin(\omega t)d(\omega t)}_{\delta_0/\delta_u<\overline{\delta}_c\leq 1}\right\} \tag{6}$$

Since $Q$ is a function of $A_{E1,1}$ and $A_{E2,1}$, which are the approximated amplitude of eigenfunctions of segment 1 of cable 1 and segment 1 of cable 2, an arbitrary scale parameter is required in Equation (6). It is assumed that the arbitrary amplitude of the eigenfunction can be expressed in terms of the cable length, i.e., $A_{Ek,j} = \kappa L_k$ [29,46], where $\kappa$ is a constant that needs to be determined based on the maximum strain of the cross-tie material. Thus, as can be observed from Equation (6), the non-dimensional equivalent linearized cross-tie axial stiffness $\overline{K}_{eq}$ depends on the cable network modal frequency $\Omega$, the cross-tie location $\varepsilon$, the frequency ratio $\eta$, the amplitude of vibration $\kappa$ and the cross-tie material property. It can be rewritten as follows:

$$\overline{K}_{eq}(\Omega,\ \varepsilon,\ \eta,\ \kappa, \text{material}) = \overline{K}_c + \frac{\delta_u}{\pi Q}\left\{\underbrace{\int_0^{2\pi} \overline{F}_c(\overline{\delta}_c)\sin(\omega t)d(\omega t)}_{\delta_0/\delta_u<\overline{\delta}_c\leq 1}\right\} \tag{7}$$

where $\overline{K}_c$ is the non-dimensional cross-tie linear axial stiffness. After obtaining $\overline{K}_{eq}$, the modal properties of a considered cable network can be determined by solving the eigenvalue problem of the equivalent linear cable network system. As shown in Figure 2, the

cross-tie divides each main cable into two segments. The equation of motion for the in-plane free vibration of the $j^{\text{th}}$ segment in the $k^{\text{th}}$ cable can be expressed as [48]

$$\frac{d^2 \tilde{v}_{k,j}}{dx_{k,j}^2} + \beta_k^2 \tilde{v}_{k,j} - \tilde{h}_k \frac{m_k g}{H_k^2} - \mu_k \beta_k^2 \frac{d^4 \tilde{v}_{k,j}}{dx_{k,j}^4} = 0 \tag{8}$$

where $\beta_k = \omega_k \sqrt{m_k / H_k}$ is the wave number and $\mu_k = E_k I_k / \left( H_k L_k^2 \right)$ is the flexural rigidity parameter of the $k^{\text{th}}$ cable. $\tilde{h}_k = \frac{8 d_k}{L_k^2 L_{ek} / (E_k A_k)} \sum_{j=1}^{2} \int_0^{l_{k,j}} \tilde{v}_{k,j}\left(x_{k,j}, t\right) dx_{k,j}$ is the additional chord tension of the $k^{\text{th}}$ cable induced by dynamic motion [49], of which $d_k = m_k g L_k^2 / (8 H_k)$ is the sag of the $k^{\text{th}}$ cable. $L_k$ and $H_k$ are the length and the chord tension of the $k^{\text{th}}$ cable ($k = 1, 2$), respectively. $L_{ke} = L_k \left[1 + 8(d_k / L_k)^2\right]$ is the effective length of the $k^{\text{th}}$ cable static profile [49], and $E_k A_k$ is the axial stiffness of the $k^{\text{th}}$ cable ($k = 1, 2$).

Assume one end of the cable segment is fixed, the general solution to Equation (8) can be expressed in a form shown in Equation (9). Derivation details of Equation (9) can be found in Younespour and Cheng [42] and Fujino and Hoang [48].

$$\tilde{v}_{k,j}\left(x_{k,j}\right) = A_{k,j}\left[\gamma_{ak}\sin(\gamma_{b\,k}x_{k,j}) - \gamma_{bk}\sinh(\gamma_{a\,k}x_{k,j})\right] + B_{k,j}\left[\cos(\gamma_{b\,k}x_{k,j}) - \cosh(\gamma_{a\,k}x_{k,j})\right]$$
$$+ \frac{8 d_k}{(\beta_k L_k)^2} \frac{\tilde{h}_k}{H_k}\left[1 - \cosh(\gamma_{a\,k}x_{k,j})\right] \tag{9}$$

where $A_{k,j}$ and $B_{k,j}$ denote the shape function constants, which can be determined from the continuity conditions at the point where the cross-tie and the main cables are connected. $\gamma_{ak}^2 = \frac{1}{2\mu_k L_k^2}\left(\sqrt{1 + 4\mu_k \beta_k^2 L_k^2} + 1\right)$ and $\gamma_{bk}^2 = \frac{1}{2\mu_k L_k^2}\left(\sqrt{1 + 4\mu_k \beta_k^2 L_k^2} - 1\right)$ represent the auxiliary wave number parameters ($k = 1, 2$).

Equation (9) finds application in characterizing the in-plane transverse displacement of any cable segment in a cable network. For the shallow-flexible-cable network shown in Figure 2, the eight unknown shape function constants $A_{k,j}$ and $B_{k,j}$ ($k = 1, 2; j = 1, 2$) can be determined through the six compatibility conditions and two equilibrium conditions established at the installation locations of the cross-tie, and they are outlined as follows [28,50]:

$$\begin{array}{ll}
\tilde{v}_{1,1}(l_{1,1}, t) = \tilde{v}_{1,2}(l_{1,2}, t), & \tilde{v}_{2,1}(l_{2,1}, t) = \tilde{v}_{2,2}(l_{2,2}, t) \\
\tilde{v}'_{1,1}(l_{1,1}, t) = -\tilde{v}'_{1,2}(l_{1,2}, t), & \tilde{v}'_{2,1}(l_{2,1}, t) = -\tilde{v}'_{2,2}(l_{2,2}, t) \\
\tilde{v}''_{1,1}(l_{1,1}, t) = \tilde{v}''_{2,1}(l_{2,1}, t), & \tilde{v}''_{2,1}(l_{2,1}, t) = \tilde{v}''_{2,2}(l_{2,2}, t)
\end{array} \tag{10}$$

$$K_{eq}\left[\tilde{v}_{2,1}(l_{2,1}, t) - \tilde{v}_{1,1}(l_{1,1}, t)\right] = E_1 I_1\left[v'''_{1,1}(l_{1,1}, t) - v'''_{1,2}(l_{1,2}, t)\right] \tag{11a}$$

$$E_1 I_1\left[v'''_{1,1}(l_{1,1}, t) - v'''_{1,2}(l_{1,2}, t)\right] = -E_2 I_2\left[v'''_{2,1}(l_{2,1}, t) - v'''_{2,2}(l_{2,2}, t)\right] \tag{11b}$$

An application of the above conditions to Equation (9) leads to eight dimensionless algebraic equations, which can be written in a matrix form as $\mathbf{G}H^T = 0$, of which the matrix $\mathbf{G}$ is a function of the dimensionless parameters $\Omega$, $\overline{K}_{eq}$, and $\kappa$, and $H = \begin{bmatrix} A_{1,1} & A_{1,2} & A_{2,1} & A_{2,2} & B_{1,1} & B_{1,2} & B_{2,1} & B_{2,2} \end{bmatrix}$ is the coefficient matrix. For the equivalent linear cable network system derived using the HBM approach, its modal properties can be obtained from the nontrivial solutions to $\mathbf{G}H^T = 0$. (For more details please refer to [42].)

## 3. Impact of Cross-Tie Nonlinearity on the Modal Behavior of Shallow-Flexible Cable Networks

This section delves into the investigation of the influence of cross-tie nonlinearity on the modal response of two-cable networks with varying configurations. The analytical model developed in the previous section is employed for this purpose. Initially, the

proposed analytical model will be applied to evaluate the impact of cross-tie nonlinearity on the in-plane dynamic response of a twin-cable network with a cross-tie located at its mid-span. Subsequently, a more generalized case is considered: a symmetric unequal-length two-cable network with a nonlinear cross-tie positioned at the quarter span. In both numerical examples, the analysis takes into account the bending stiffness and sag of the main cables. The target cable is assumed to be the same as one of the stay cables on the Tatara Bridge [51], which has the following properties: $L_1 = 260.20$ m, $H_1 = 4689$ kN, $m_1 = 91.6$ kg/m, $E_1 I_1 = 2231$ kN·m². The properties of the neighboring cable (cable 2) are the same as those of the target cable, except that in the symmetric unequal-length cable network example, it has a shorter length of $L_2 = 234$ m. If we define $\rho = L_2/L_1$ as the length ratio between the target cable and the neighboring one, it will be $\rho = 0.9$. In addition, since the considered cables are shallow flexible cables, the cable sag is taken into account and it is assumed to have a value of $\lambda_1^2 = 3$, where $\lambda_k^2$ ($k = 1, 2$) is the $k^{\text{th}}$ cable sag parameter defined as $\lambda_k^2 = \left(\frac{8d_k}{L_k}\right)^2 \frac{L_k}{H_k L_{ke}/E_k A_k}$ [49]. It is worth mentioning that on real cable-stayed bridges, the flexural rigidity parameter $\mu$ and the Irvine parameter $\lambda^2$ for stay cables of 20 m to 300 m in length are within the ranges of $2.82 \times 10^{-6} < \mu < 1.59 \times 10^{-2}$ and $0.6 < \lambda^2 < 3$, respectively [51].

To verify the validity of the proposed analytical model, finite element (FE) simulations will be conducted to compare the numerical results with those obtained from the proposed analytical model. In addition, to assess the influence of cross-tie material nonlinearity on the in-plane modal response of the two-cable networks in the numerical examples, their modal behavior is also analyzed with the original nonlinear cross-tie being replaced by a linear cross-tie of $K_c = 2500$ kN/m. A comparison is made between the nonlinear cross-tie case and the linear one.

### 3.1. FE Simulation

For the validation of the proposed analytical model, a finite element simulation is carried out using ABAQUS software. In this developed FE model, the B21 beam element is chosen to replicate the in-plane behavior of the main cables. The B21 beam element is a one-dimensional Timoshenko beam element characterized by two nodes. Each node encompasses three degrees of freedom: two for translations along the x- and y-axes and one for rotation about the z-axis. Employing a mesh size of 1 m, the result is a configuration of 260 elements and 261 nodes for each cable. Notably, all three degrees of freedom of the nodes located at the two ends of cable model are constrained, resulting in a fixed-fixed boundary condition that aligns with the analytical model. To account for the influence of cable pretension, the initial stress is incorporated within the B21 beam elements. Furthermore, a uniformly distributed load is exerted vertically downward along the cable length to simulate the effect of the cable self-weight. In the FE model, the T2D2 truss element is harnessed to simulate the nonlinear behavior of the cross-tie. This element has two nodes, with two degrees-of-freedom associated with each node. The cross-tie material is assumed to be an elastoplastic type with a stress–strain curve that exhibits a strain-softening behavior. In addition, the linear cross-tie is simulated using the SPRING2 element.

In cases where the cross-tie exhibits linear behavior, the cable network's modal characteristics can be established by solving an eigenvalue problem utilizing a linear stiffness matrix. Nonetheless, if the cross-tie material behaves elastoplastically; the stiffness matrix of cable network is no longer linear and necessitates continuous updates based on cross-tie deformation. Consequently, the modal properties cannot be ascertained through the conventional eigenvalue problem approach. In this study, the modal frequencies of the examined cable network featuring a nonlinear cross-tie are determined by applying the Fast Fourier Transform (FFT) to the displacement response of the analyzed system. For this purpose, the target cable is displaced transversely at five different locations along its span and then released suddenly to excite free vibration of the entire cable network. The introduced initial displacement is applied at $1/5L_1$, $1/3L_1$, $1/2L_1$, $2/3L_1$, and $3/4L_1$ along the target cable, where $L_1$ is the length of the target cable. The maximum axial strain in

the cross-tie is defined based on the parameter $\kappa$ introduced in the analytical model, which is taken as $\kappa = 0.0003$. The free vibration response, in terms of the target cable mid-span displacement time history, is recorded for 40 s. The Fast Fourier Transform (FFT) is then employed to analyze this recorded dynamic response, aiming to discern the modal frequencies inherent to the investigated cable network. The modal frequencies corresponding to the initial 10 modes of both the analyzed twin-cable network and the unequal-length two-cable network, as presented in the two numerical examples, are listed in Tables 1 and 2 from the FE simulation, respectively.

**Table 1.** In-plane modal properties of a twin-cable network with either a linear or a nonlinear cross-tie at the mid-span.

| Mode Number | Modal Frequency (Hz) | | | Diff. * (%) | Mode Shape | |
| --- | --- | --- | --- | --- | --- | --- |
| | Nonlinear Cross-Tie | | Linear Cross-Tie | | Nonlinear Cross-Tie | LinearCross-Tie |
| | Analytical | FEA | Analytical | | | |
| 1 | 0.4649 | 0.4587 | 0.4649 | 1.3 | GM, 1-Sym., IP | GM, 1-Sym., IP |
| 2 | 0.4921 | 0.4874 | 0.8733 | 1.0 | GM, 1-Sym., OP | GM, 1-Asym., IP |
| 3 | 0.8654 | 0.8599 | 0.8802 | 0.6 | GM, 1-Asym., OP | LM-RS |
| 4 | 0.8733 | 0.8666 | 0.8802 | 0.8 | GM, 1-Asym., IP | LM-LS |
| 5 | 1.3540 | 1.3421 | 1.3540 | 0.9 | GM, 2-Sym., IP | GM, 2-Sym., IP |
| 6 | 1.3856 | 1.3789 | 1.7836 | 0.5 | GM, 2-Sym., OP | GM, 2-Asym.,IP |
| 7 | 1.4361 | 1.4301 | 1.7955 | 0.4 | GM, 2-Asym., OP | LM-RS |
| 8 | 1.7836 | 1.7734 | 1.7955 | 0.6 | GM, 2-Asym., IP | LM-LS |
| 9 | 2.3342 | 2.3409 | 2.3342 | 0.3 | GM, 3-Sym., IP | GM, 3-Sym., IP |
| 10 | 2.4901 | 2.4840 | 2.7454 | 0.2 | GM, 3-Sym., OP | GM, 3-Asym., IP |

\* Absolute percentage difference between the analytically and numerically obtained modal frequencies considering cross-tie nonlinearity (GM: global mode, LM: local mode, Sym: symmetric, Asym: anti-symmetric, FEA: finite element analysis, IP: in-phase, OP: out-of-phase).

**Table 2.** In-plane modal properties of an unequal-length two-cable network with either a linear or a nonlinear cross-tie at the quarter span.

| Mode Number | Modal Frequency (Hz) | | | Diff. * (%) | Mode Shape | |
| --- | --- | --- | --- | --- | --- | --- |
| | Nonlinear Cross-Tie | | Linear Cross-Tie | | Nonlinear Cross-Tie | Linear Cross-Tie |
| | Analytical | FEA | Analytical | | | |
| 1 | 0.4823 | 0.4790 | 0.4894 | 0.7 | GM, 1-PS, IP | GM, 1-PS, IP |
| 2 | 0.5998 | 0.5900 | 0.6331 | 1.6 | GM, 1-PS, OP | LM-RS, OP |
| 3 | 0.9682 | 0.9591 | 0.9781 | 0.9 | GM, 1-PAS, IP | GM, 1-PAS, IP |
| 4 | 1.2091 | 1.1957 | 1.2590 | 1.1 | GM, 1-PAS, OP | LM-RS, OP |
| 5 | 1.4489 | 1.4396 | 1.4657 | 0.6 | LM, Cable 1, 2-PS | GM, 2-PS, IP |
| 6 | 1.8230 | 1.8097 | 1.8491 | 0.7 | LM, Cable 2, 2-PS | GM, 2-PAS, OP |
| 7 | 1.9133 | 1.9011 | 1.9515 | 0.6 | LM, Cable 1, 2-PAS | LM, OP |
| 8 | 2.1444 | 2.1379 | 2.1766 | 0.3 | LM, Cable 2, 2-PAS | LM, OP |
| 9 | 2.3890 | 2.3784 | 2.4315 | 0.4 | LM, Cable 1, 3-PS | LM, OP |
| 10 | 2.5622 | 2.5486 | 2.5676 | 0.5 | LM, Cable 2, 3-PS | LM, OP |

\* Absolute percentage difference between the analytically and numerically obtained modal frequencies considering cross-tie nonlinearity, (GM: global mode, LM: local mode, PS: Pseudo symmetric, PAS: Pseudo anti-symmetric, FEA: finite element analysis, IP: in-phase, OP: out-of-phase).

### 3.2. Twin-Cable Network with a Nonlinear Cross-Tie at the Mid-Span

To evaluate the influence of cross-tie nonlinearity on the modal response of shallow-flexible-cable networks, a nonlinear cross-tie with an elastoplastic behavior given in Figure 3 is considered. The total non-dimensional elastic deformation of the cross-tie is assumed to be $\delta_0/\delta_u = 0.031$, which means that the eigenfunction amplitude $A_E$ of the equivalent linearized cable network system is generally small in comparison to the cable length. In this study, the cross-tie is 8 m long and $\delta_u = 0.1$. Therefore, the maximum vibration amplitude can be 0.8 m, which leads to $A_E$ being less than $L/300$. To apply the HBM to determine the equivalent linear cross-tie stiffness, the non-dimensional force-displacement relation of the cross-tie material in Figure 3 needs to be fitted in the form of a piecewise power series polynomial defined in Equation (1). An $R^2$ value of 95% is considered to be an acceptable accuracy for curve fitting. Based on this criterion, a 5th order polynomial is selected, the $R^2$ value of which is 95.3%. The fitted curve is given in Equation (12) and portrayed in Figure 3. The arbitrary amplitude constant $\kappa$ is assumed to be 0.0003, which yields a strain of 0.019 in the cross-tie.

$$\overline{F}_c(\overline{\delta}_c) = \begin{cases} 51\overline{\delta}_c & 0 < \overline{\delta}_c \leq 0.031 \\ \\ 12.33\overline{\delta}_c{}^5 - 31.99\overline{\delta}_c{}^4 + 31.06\overline{\delta}_c{}^3 - 14.17\overline{\delta}_c{}^2 + 3.36\overline{\delta}_c + 0.504 & 0.031 < \overline{\delta}_c \leq 1 \end{cases} \tag{12}$$

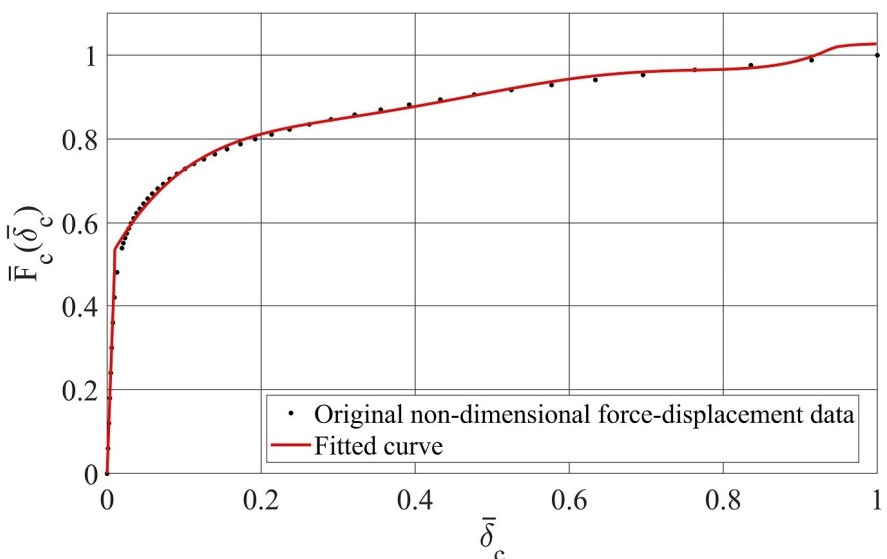

**Figure 3.** Fitted curve to the original non-dimensional force-displacement data of the cross-tie material (strain-softening behavior).

Table 1 presents the modal frequencies of the first 10 modes of the studied twin-cable network obtained from the proposed analytical model. As a comparison, the modal response of the twin-cable network is also analyzed by replacing the nonlinear cross-tie with a linear one having $K_c = 2500$ kN/m. This set of results is also shown in Table 1. The corresponding mode shapes in both the linear and the nonlinear cross-tie cases are depicted in Figure 4.

The modal frequencies of the twin-cable network predicted by the FE simulation and those obtained using the proposed analytical model demonstrate good agreement. As can be seen in Table 1, the maximum percentage difference is only 1.3%, which occurs in Mode 1. This verifies the validity of the proposed analytical model, of which the HBM is applied to determine the equivalent linear axial stiffness of the cross-tie in the studied nonlinear cable network system.

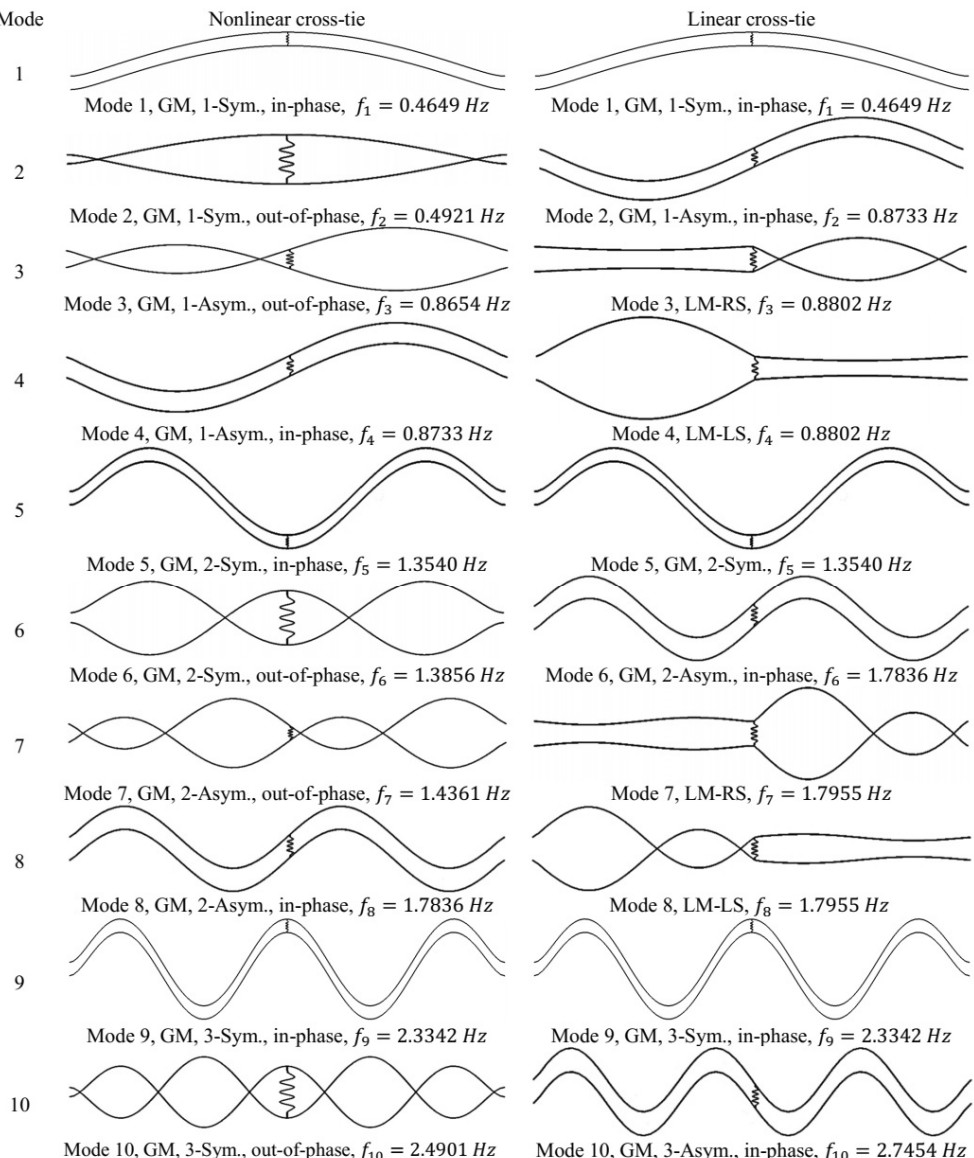

**Figure 4.** Comparison of the modal response of the first 10 modes of a twin-cable network with either a linear or a nonlinear cross-tie (GM: global mode; LM: local mode; Sym: symmetric; Asym: anti-symmetric).

A comparison of the modal responses between the nonlinear cross-tie case and the linear one reveals that the presence of cross-tie nonlinearity has no impact on the modal frequencies and modal order of all symmetric in-phase global modes. This is consistent with the results reported in earlier studies [28,42,52], in which it was shown that the modal properties of these modes in a twin-cable network are unaffected by the change in the cross-tie property. This happens due to the fact that in the symmetric in-phase global modes of a twin-cable network, each cable vibrates independently, and the frequency of the cable network global modes remains the same as that of a single cable. However, although the modal frequencies of the anti-symmetric in-phase global modes remain unchanged, they are observed to delay to higher order modes. For example, by considering nonlinear cross-tie behavior, the modal order of the first and the second anti-symmetric in-phase global modes are respectively delayed from Mode 2 to Mode 4 and from Mode 6 to Mode 8. In addition, it is seen that by replacing a linear cross-tie with a nonlinear one, the out-of-phase global modes are excited. As illustrated in Figure 4, when a cross-tie exhibits nonlinear behavior, all the local segment modes evolved into anti-symmetric out-of-phase global modes with

lower modal frequencies. For instance, the local segment modes in Mode 3 and Mode 4 of the linear cross-tie case evolve into the first anti-symmetric out-of-phase global mode in the case of the nonlinear cross-tie, which is Mode 3. In addition, it is noticed that in the case of the linear cross-tie, there is no symmetric out-of-phase global mode. However, this type of mode is excited when the cross-tie manifests nonlinear behavior (e.g., Modes 2, 6, and 10). Results in Table 1 and Figure 5 clearly indicate that within the first 10 cable network modes, while four of them are local modes in the linear cross-tie case, no local mode appears in the nonlinear cross-tie case. The adoption of a nonlinear cross-tie would render them to evolve into global modes and thus delay the formation of local modes to a high order. It is worth mentioning that the reduction in the rigidity of the network and thus the system frequency decrement is due to the use of a cross-tie material with strain-softening behavior.

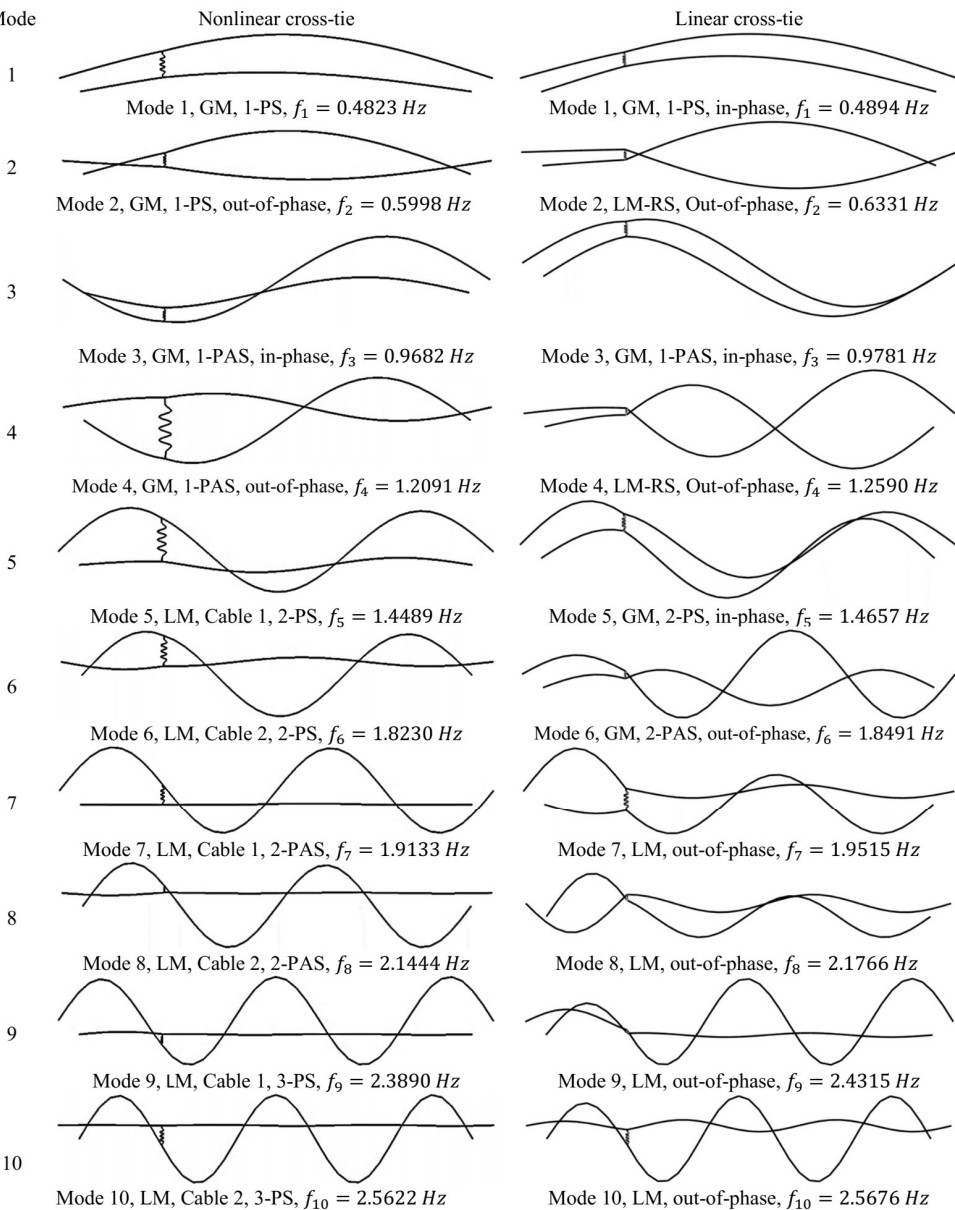

**Figure 5.** Comparison of the modal response of first 10 modes of an unequal-length two-cable network with either a linear or a nonlinear cross-tie (GM: global mode; LM: local mode; PS: pseudo symmetric; PAS: pseudo anti-symmetric).

*3.3. Symmetric Unequal-Length Two-Cable Network with a Nonlinear Cross-Tie at a Quarter-Span*

As an illustration of a more general example, this section examines the modal behavior of a symmetric unequal-length two-cable network with a nonlinear cross-tie placed at the quarter span (i.e., $\varepsilon_{k,1} = 1/4$ and $\varepsilon_{k,2} = 3/4$). In this setup, the two constituent cables are assumed to possess the same properties as the main cables outlined in the first example, except the neighboring cable has a shorter length of $L_2 = 234$ m. The horizontal offset on both ends of cable 2 are the same, i.e., $O_L = O_R = 13$ m. The same nonlinear restoring force model used in the first example is considered, and the arbitrary amplitude constant $\kappa$ is assumed to be 0.0003. Similarly, modal analysis has also been conducted for the linear cross-tie case with $K_c = 2500$ kN/m. Table 2 presents the modal characteristics of the initial 10 modes within the investigated two-cable network, encompassing both the instances of a nonlinear cross-tie and a linear one. These respective mode shapes are illustrated in Figure 5.

As can be seen from Figure 5, when the cross-tie behaves linearly, there are four global modes and six local modes in the first 10 modes with alternating modal orders. On the other hand, when the cross-tie exhibits nonlinear behavior, although there are still four global modes and six local modes among the first 10 modes, all four global modes are excited at a low order from Mode 1 to Mode 4, while all six local modes appear at a higher order. This implies that the presence of material nonlinearity in the cross-tie could delay the formation of local modes. Compared to the linear cross-tie case, when the cross-tie behavior is nonlinear, its axial stiffness would be decreased, resulting in less constrain to the cable transverse motion at their connection point and allowing all connected cable segments to oscillate simultaneously, which would encourage the development of global modes and suppress the formation of local modes. This could be one of the reasons why low-order modes in cable networks on real cable-stayed bridges are often observed to be global modes [26]. There are four distinct changes in the characteristics of the cable network modal response when the cross-tie behavior changes from linear to nonlinear: (a) the modal order of the lowest in-phase global modes, both pseudo symmetric (Mode 1) and pseudo anti-symmetric (Mode 3), are not affected. However, while their mode shapes remain more or less the same, the modal frequencies drop slightly due to the strain-softening behavior of the nonlinear cross-tie. (b) The out-of-phase local segment modes evolve into out-of-phase global modes with a drop in the modal frequency. This occurs in Mode 2 and Mode 4. In the case of Mode 2, it is a local right segment mode in the linear cross-tie case, of which the right segment of cable 1 and cable 2 vibrates out-of-phase in the first pseudo symmetric pattern. With the cross-tie exhibiting nonlinear behavior, the oscillation is extended to the left segment of both cables and thus excites the first pseudo symmetric out-of-phase global mode. Similarly, such a change in the cross-tie behavior results in the first pseudo anti-symmetric out-of-phase global mode being evolved from an out-of-phase local right segment mode in Mode 4. (c) The evolution of a global mode to a local mode is dominated by one of the main cables. This type of mode shape change is observed in Mode 5 and 6, where a second-order pseudo symmetric in-phase global mode becomes a local mode dominated by the oscillation of cable 1 in a second-order pseudo symmetric pattern (Mode 5), and a second-order pseudo anti-symmetric out-of-phase mode turns into a local mode of cable 2 in the second-order pseudo symmetric shape (Mode 6). (d) A local mode dominated by the oscillations of some cable segments becomes a local mode dominated by the motion of one of the main cables. Modes 7, 8, 9, and 10 all fall into this category. The above observations clearly indicate that the presence of cross-tie nonlinearity would not only affect the modal frequencies and modal order, but could also considerably alter the associated mode shapes and thus would significantly affect the in-plane modal response of a cable network.

## 4. Parametric Study

Results of numerical examples in Section 3 show that the presence of cross-tie material nonlinearity could considerably change the modal behavior of a cable network, and the

prediction of the modal response of a cable network with a nonlinear cross-tie would be influenced by the approximation of $\overline{K}_{eq}$, which is the non-dimensional equivalent linear axial stiffness of the cross-tie. The analytical model of a cable network with a nonlinear cross-tie proposed in Section 2 shows that $\overline{K}_{eq}$ is not only mode-dependent but also influenced by a number of parameters, including the cross-tie material property, the cable vibration amplitude represented by $\kappa$, the cross-tie position $\varepsilon$, and the length ratio $\rho$. To fully understand the impact of different parameters on the equivalent linear stiffness of the cross-tie and the fundamental frequency of the studied cable network, a parametric study is carried out using the proposed analytical model.

*4.1. Effect of the Cross-Tie Material Property*

The cross-tie material property has an important role in the dynamic behavior of cable networks. To identify its effect on the fundamental frequency of the cable network, $\Omega_1$, and the cross-tie equivalent linear stiffness associated with the network fundamental mode, $\overline{K}_{eq_1}$, two different types of cross-tie material nonlinearity, strain softening and strain hardening, are considered. They are assumed to have the same behavior in the linear range. The material behavior of these two types of nonlinearity are illustrated in Figures 3 and 6, respectively. The associated non-dimensional force-displacement relations are fitted by piecewise power series polynomials of order 5 and 4, with $R^2$ being 95.3% and 97.3%, respectively. The fitted polynomials are given in Equations (12) and (13), respectively. The total non-dimensional elastic deformation of the strain-hardening material is assumed to be $\delta_0/\delta_u = 0.042$.

$$\overline{F}_c(\overline{\delta}_c) = \begin{cases} 31.5\overline{\delta}_c & 0 < \overline{\delta}_c \leq 0.042 \\ \\ -0.45\overline{\delta}_c^{\,4} + 1.69\overline{\delta}_c^{\,3} - 1.43\overline{\delta}_c^{\,2} + 0.72\overline{\delta}_c + 0.45 & 0.042 < \overline{\delta}_c \leq 1 \end{cases} \tag{13}$$

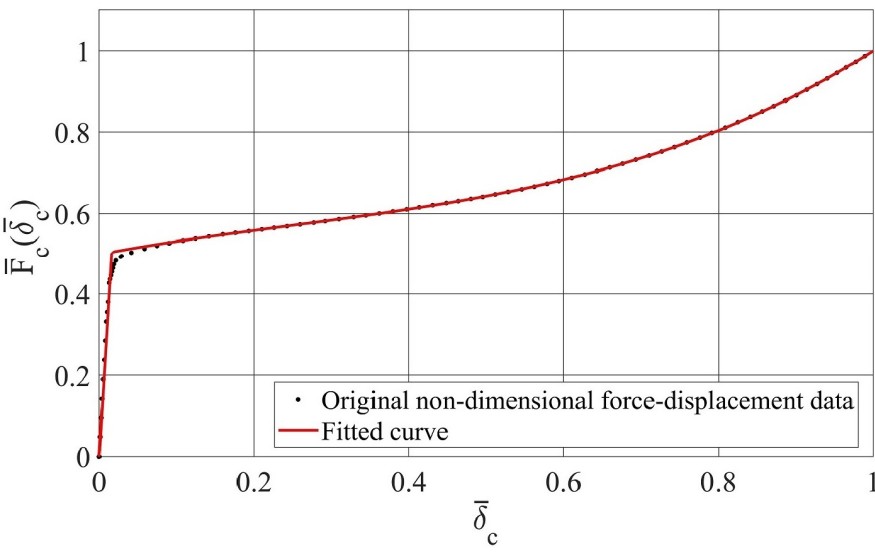

**Figure 6.** Fitted curve to the original non-dimensional force-displacement data of the cross-tie material (strain-hardening behavior).

To analyze the cross-tie material nonlinearity effect on $\overline{K}_{eq_1}$ and $\Omega_1$, the two cable networks studied in the previous numerical examples are considered, except now, the cross-tie in both networks is assumed to be located at the main cable mid-span ($\varepsilon = 0.5$). Irvine's parameter $\lambda^2$ and the flexural rigidity parameter $\mu$ of the main cables are taken to be 3 and $7.02 \times 10^{-6}$, respectively.

Figure 7a illustrates how the cable network fundamental frequency would be affected by the cross-tie material type. In the case of a twin-cable network, since the global modes would not be affected by the cross-tie properties [28,42,52], the change in the cross-tie material type and deformation would thus have no influence on the fundamental frequency. This, as reflected in Figure 7a, in that $\Omega_1$ remains as a constant when $\bar{\delta}_c$ changes, and the $\Omega_1/\pi - \bar{\delta}_c$ curves associated with the two different cross-tie material types overlap with each other. On the other hand, in an unequal-length cable network, when the cross-tie deformation exceeds the elastic limit, $\Omega_1$ is observed to decrease as the cross-tie experiences more deformation, except when the cross-tie material is of the strain-hardening type, for which the fundamental frequency would gradually recover once the deformation of the cross-tie is over a characteristic value corresponding to the curvature change in the $\bar{F}_c - \bar{\delta}_c$ curve shown in Figure 6. For the strain-hardening material used in the current study, this characteristic value is $\bar{\delta}_c = 0.65$.

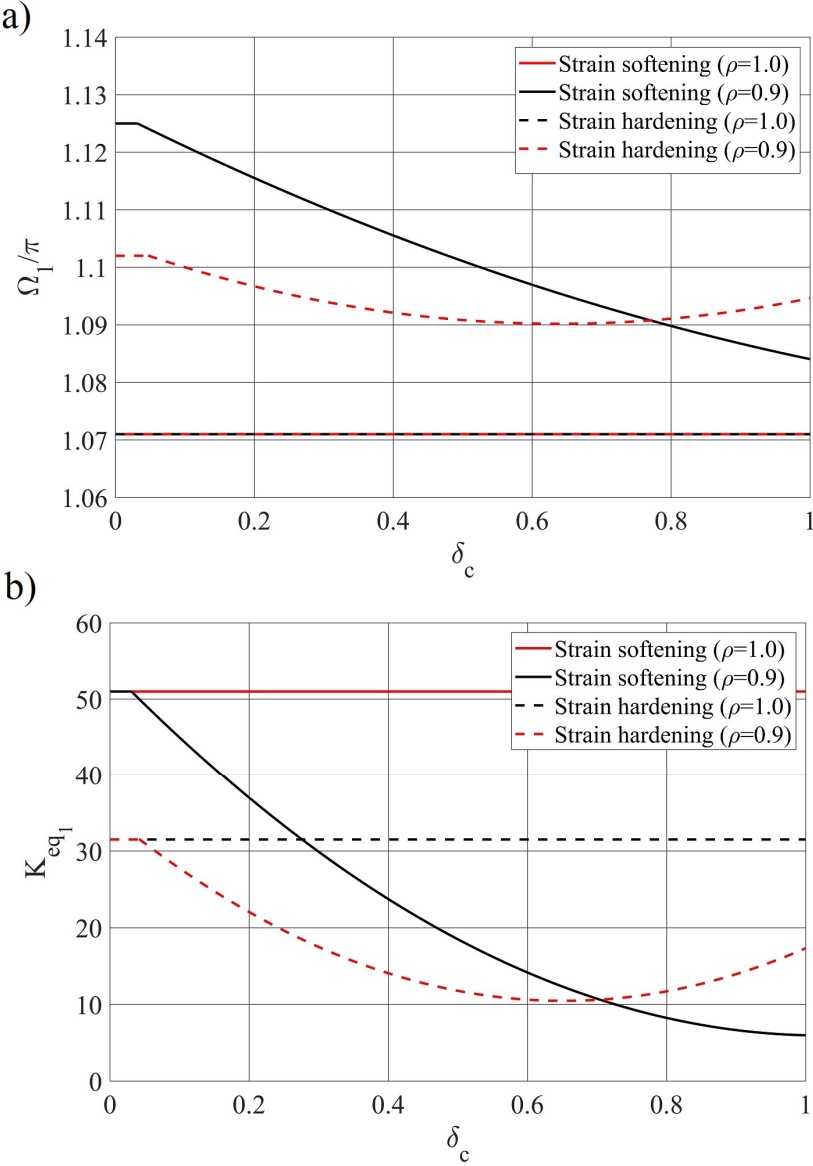

**Figure 7.** Effect of the cross-tie material nonlinearity type on different system parameters ($\varepsilon = 0.5$): (**a**) non-dimensional fundamental frequency of the cable network; (**b**) non-dimensional equivalent linear stiffness of the cross-tie.

The impact of the cross-tie material nonlinearity type on its equivalent linear stiffness $\bar{K}_{eq_1}$ is shown in Figure 7b. Since the cross-tie equivalent linear stiffness depends on the

modal frequency, the patterns of the $\overline{K}_{eq_1} - \bar{\delta}_c$ curves are consistent with the respective $\Omega_1/\pi - \bar{\delta}_c$ curves in Figure 7a, i.e., those of the twin-cable network remain as horizontal lines, whereas those of the strain-softening and strain-hardening material types manifest as a monotonically decreasing trend and a decreasing-recover pattern, respectively. In addition, for the two types of nonlinear cross-tie material properties selected in this study, the strain-softening type of the cross-tie material is observed to render $\overline{K}_{eq_1}$ to be more sensitive to cross-tie deformation.

As indicated in Equation (8), the equivalent linear stiffness of the cross-tie, $\overline{K}_{eq}$, depends on the vibration amplitude parameter $\kappa$. It is noteworthy that the non-dimensional cross-tie deformation $\bar{\delta}_c$ is directly related to $\kappa$, so the influence of $\kappa$ on $\Omega_1$ and $\overline{K}_{eq}$ would be similar as that of $\bar{\delta}_c$ shown in Figure 7a,b.

*4.2. Effect of the Cross-Tie Position*

This section investigates the effect of the cross-tie position $\varepsilon$ on its equivalent linear stiffness, $\overline{K}_{eq_1}$, and the cable network fundamental frequency, $\Omega_1$. As mentioned earlier, the fundamental frequency of a twin-cable network is independent of the cross-tie position and material properties. Therefore, the cross-tie position effect is examined in this section using an unequal-length cable network with $\rho = 0.9$. Two different vibration amplitude parameters, i.e., $\kappa = 0.001$ and $\kappa = 0.0015$, are considered, and the position of the cross-tie varies between $\varepsilon = 0.2$ and $\varepsilon = 0.8$. Figure 8a,b portray the variation of $\overline{K}_{eq_1}$ and $\Omega_1/\pi$, respectively, with respect to the cross-tie position when the cross-tie material has either strain-softening or strain-hardening behavior. Since the considered unequal-length cable network has a symmetric layout, the $\overline{K}_{eq_1} - \varepsilon$ and $\Omega_1/\pi - \varepsilon$ curves for both material types show a symmetric pattern. Upon moving the cross-tie from the cable end to the mid-span, $\Omega_1$ and $\overline{K}_{eq_1}$ decrease gradually until they reach their minimum values at $\varepsilon = 0.5$. This pattern can be explained by referring to Figure 9, where the fundamental mode shape of the studied cable network is shown schematically. In the figure, "a" and "b" represent cross-tie locations closer to the cable end and closer to the cable mid-span, respectively. As can be seen from the figure, the axial deformation of the cross-tie increases as its installation location moves towards the cable mid-span. When the deformation exceeds the elastic range of the cross-tie material, the axial stiffness of the cross-tie would be reduced considerably, as shown in Figures 3 and 6. Therefore, the in-plane stiffness of the entire cable network would be less, leading to a lower fundamental frequency, $\Omega_1$, and a smaller equivalent linear stiffness, $\overline{K}_{eq_1}$. For example, when $\kappa = 0.001$, by changing the cross-tie position from $\varepsilon = 0.2$ to $\varepsilon = 0.5$, the non-dimensional fundamental frequency of the cable network, $\Omega_1$, decreases from $1.107\pi$ to $1.094\pi$ by 1.2%, and the non-dimensional equivalent linear stiffness of the cross-tie for the fundamental mode, $\overline{K}_{eq_1}$, drops 33.7% from 18.62 to 12.34. It is worth pointing out that this finding contradicts the results reported in earlier studies based on a linear cross-tie [52], where it was found that moving the cross-tie to the cable mid-span would enhance the in-plane stiffness of a cable network and thus increase its fundamental frequency. This contradiction suggests that neglecting the cross-tie nonlinear behavior can result in a misleading prediction of the cable network dynamic response. Therefore, to evaluate the cable network dynamic behavior more accurately, the cross-tie nonlinearity needs to be considered in the analysis. Nevertheless, it is noteworthy that from a practical point of view, such an impact is minor.

Further, a comparison between Figure 8a and b indicates that the influence of the cross-tie position on $\Omega_1$ and $\overline{K}_{eq_1}$ would not only be affected by the main cable vibration amplitude, but also depend on the cross-tie material nonlinearity type. When a more severe cable vibration occurs, while $\Omega_1$ and $\overline{K}_{eq_1}$ would be less sensitive to the cross-tie position effect, should the cross-tie material be of the strain-softening type, they are affected more by the change in $\varepsilon$ if a strain-hardening-type cross-tie is used. In addition, it is noticed in Figure 8b that the cable network would have a higher fundamental frequency and a larger cross-tie equivalent linear stiffness when more severe cable vibration occurs. This is believed to result from the strain-hardening characteristic of the cross-tie material. The fact

that the axial stiffness of a cross-tie has an important contribution to the in-pane stiffness of a cable network suggests that any variation in the equivalent linear stiffness of a cross-tie will lead to a change in the system frequency. Though the equivalent linear axial stiffness of a cross-tie depends on its material type and cable vibration amplitude and its contribution to the in-plane stiffness of a cable network is also affected by the cross-tie installation location, for a given cross-tie material type and installation location, an approximate linear relationship is observed to exist between the equivalent linear axial stiffness of the cross-tie and the cable network fundamental frequency at a specific level of the cable vibration amplitude, as illustrated in Figure 8c,d.

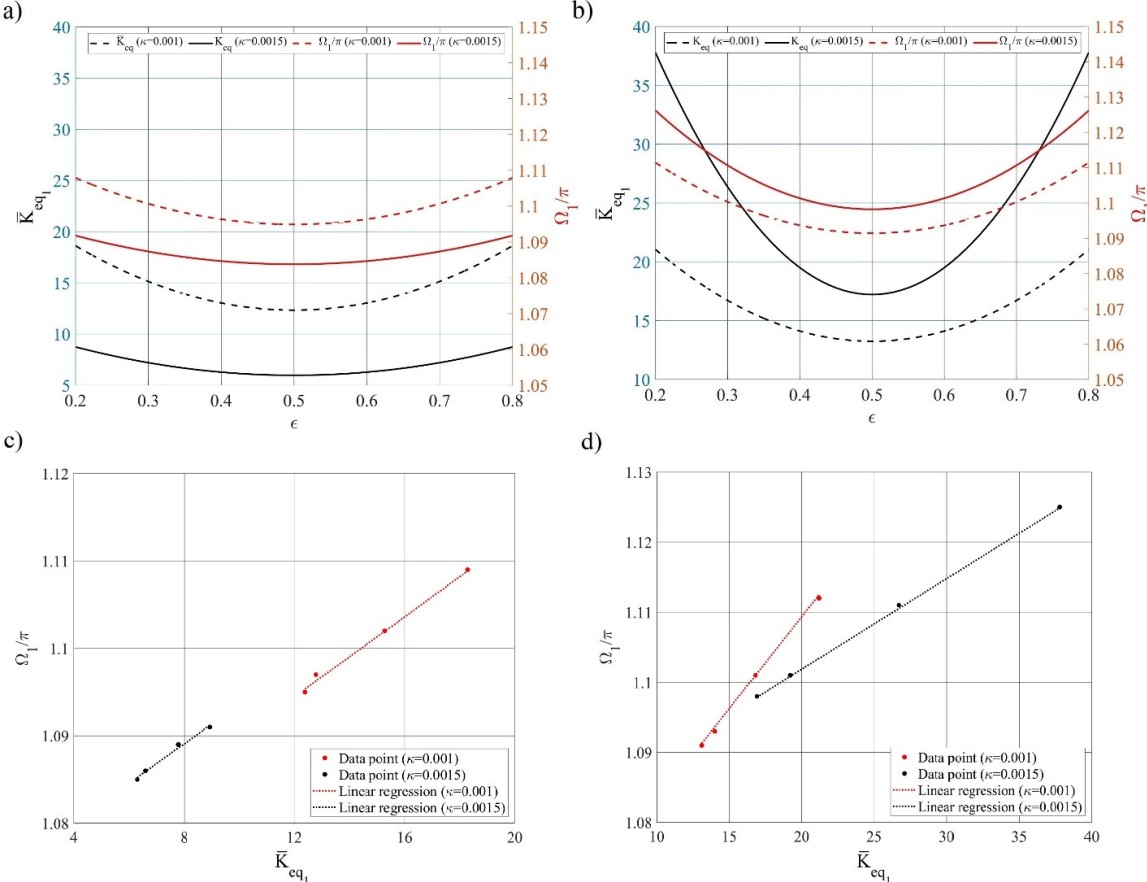

**Figure 8.** Effect of the cross-tie position on the non-dimensional equivalent linear stiffness of the cross-tie and non-dimensional fundamental frequency of the cable network: (**a**) strain-softening cross-tie behavior; (**b**) strain-hardening cross-tie behavior; (**c**) relationship between the system fundamental frequency and cross-tie equivalent linear stiffness for strain-softening-type material; (**d**) relationship between the system fundamental frequency and cross-tie equivalent linear stiffness for strain-hardening-type material.

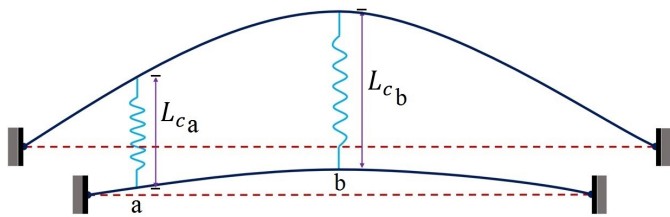

**Figure 9.** Effect of the cross-tie position on the degree of the cross-tie deformation for the first mode of vibration.

### 4.3. Effect of the Cable Length Ratio

In this section, the impact of the length ratio $\rho$ between the main cables, i.e., $\rho = L_2/L_1$, on $\Omega_1$ and $\overline{K}_{eq_1}$ will be evaluated. From the stay cable database [51], it is found that the practical range of the length ratio between two neighboring cables on real cable-stayed bridges is from 0.64 to 1.56. Figure 10 shows, schematically, the fundamental mode shape of an unequal-length two-cable network when the target cable is longer, equal to, and shorter than the neighboring cable, respectively. As defined in Figure 2, in the current study, the upper cable (main cable 1) is assumed to be the target cable, whereas the lower one is the neighboring cable. It can be seen from the figure that when $\rho > 1$, the cross-tie would be in compression. This could result in slackening of the cross-tie and would thus lead to its malfunction. In addition, as reported in [28,42], since in the in-phase global mode of a twin-cable network ($\rho = 1.0$), the two cables oscillate independently, there will be no axial deformation in the cross-tie. Based on these, a length ratio range of 0.6 to 1.0 is chosen for the current study.

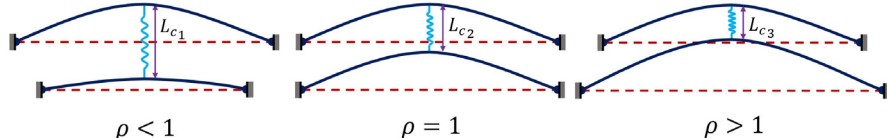

**Figure 10.** Effect of the length ratio on the cross-tie deformation in the fundamental mode of a two-cable network.

To better observe the length ratio effect, the variations in $\overline{K}_{eq_1}$ against the length ratio $\rho$ are illustrated for the strain-softening and the strain-hardening cross-tie behavior in Figure 11a,b, respectively. Three different cable vibration amplitude parameters of $\kappa = 0.0005, 0.0010$, and $0.0015$ are considered in the analysis. Results show that increasing $\rho$ would lead to a larger $\overline{K}_{eq_1}$, indicating that connecting a target cable with a longer neighboring cable would enhance the equivalent linear stiffness of the cross-tie. When the cross-tie has strain-softening behavior and the length ratio is less than a certain value, it can be observed in Figure 11a that $\overline{K}_{eq_1}$ would become less sensitive to the variation in $\rho$ when the main cables vibrate at a larger amplitude. If a 2% difference in $\overline{K}_{eq_1}$ due to the change in $\rho$ is considered as the criterion, this threshold value $\rho_t$ is identified to be 0.76 and 0.85 for $\kappa = 0.001$ and $0.0015$, respectively. Within the considered length ratio range ($0.6 \leq \rho < 1$), no clear $\rho_t$ is observed when $\kappa = 0.0005$. Further, results show that beyond $\rho_t$, $\overline{K}_{eq_1}$ rises drastically with the increase in $\rho$ for all considered $\kappa$ values. For example, as depicted in Figure 11a, when $\kappa = 0.0015$, by increasing the length ratio from 0.8 to 0.991, $\overline{K}_{eq_1}$ increases by 1217% from 3.87 to 51, while a variation of $\rho$ between 0.6 to 0.7 would lead to a change in $\overline{K}_{eq_1}$ from 3.53 to 3.56 by only 1%. As shown in Figure 10, if the target cable is connected to a longer neighboring cable and the cable network vibrates in its fundamental mode, the axial deformation of the cross-tie would be less. When $\rho$ is greater than a certain value, the cross-tie deformation would be in the elastic range, and the cross-tie stiffness is its linear stiffness, which, by referring to Figure 3, is seen to be much larger than the stiffness in the plastic zone. The same $\overline{K}_{eq_1} - \rho$ pattern is observed in Figure 11b when the cross-tie material has strain-hardening behavior, except no clear threshold of $\rho$ can be identified for the three considered levels of the vibration amplitude. Based on the pattern of the three $\overline{K}_{eq_1} - \rho$ curves portrayed in Figure 11b, it seems that $\rho_t$ in these cases is advanced to a length ratio less than 0.6. Further, a comparison between the strain-softening and strain-hardening cases reveals that while $\overline{K}_{eq_1}$ is larger when $\kappa = 0.001$ in the former case, it is larger when $\kappa = 0.0015$ in the latter. This occurs because, as mentioned in Section 4.1, in the case of strain-hardening behavior, after reaching a certain level of cross-tie deformation, the $\overline{K}_{eq_1} - \bar{\delta}_c$ curve tends to rise and partially recovers the cross-tie stiffness loss. It is important to note that the influence of the cross-tie deformation $\bar{\delta}_c$ on $\Omega_1$ and $\overline{K}_{eq}$ would be similar to that of $\kappa$.

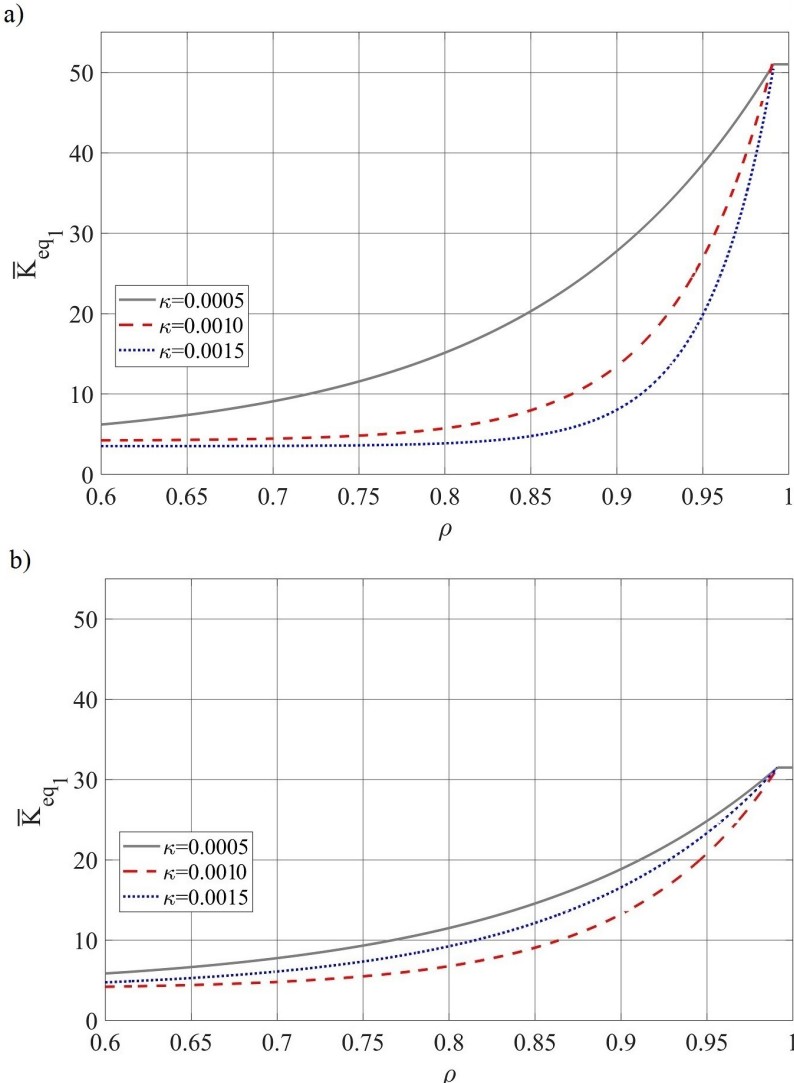

**Figure 11.** Effect of the length ratio on the non-dimensional equivalent linear stiffness $\overline{K}_{eq_1}$ of the cross-tie ($\varepsilon = 0.5$): (**a**) strain-softening cross-tie behavior; (**b**) strain-hardening cross-tie behavior.

The length ratio effect on the fundamental frequency of the studied cable network is illustrated in Figure 12, where $\Omega_1$ is observed to decrease with the increase in $\rho$ for all three considered vibration amplitude levels. Further, it is noticed that the cable network fundamental frequency converges to $\Omega_1 = 1.07\pi$ for different $\kappa$ values and material nonlinearity types at $\rho = 1$. This confirms that the fundamental frequency of a twin-cable network will not be affected by the cross-tie properties, and it remains unchanged when the cross-tie nonlinearity is taken into account. It is observed from Figures 11 and 12 that while using a longer neighboring cable would improve the equivalent linear stiffness of the cross-tie, the in-plane stiffness of the cable network would be reduced. It is worth noting that the additional in-plane stiffness provided by the network system to the target cable is the sum of the cross-tie axial stiffness and the neighboring cable lateral stiffness. With the increase in the length ratio, while a longer neighboring cable would reduce the in-plane stiffness of the cable network, a higher cross-tie axial stiffness would enhance the rigidity of the system. This suggests that the change in the network in-plane stiffness and frequency would be dictated by the net effect of the two, and a change in the cross-tie equivalent linear stiffness due to its nonlinear behavior would not necessarily result in a change in the actual in-plane stiffness of the entire cable network system with the same trend.

a)

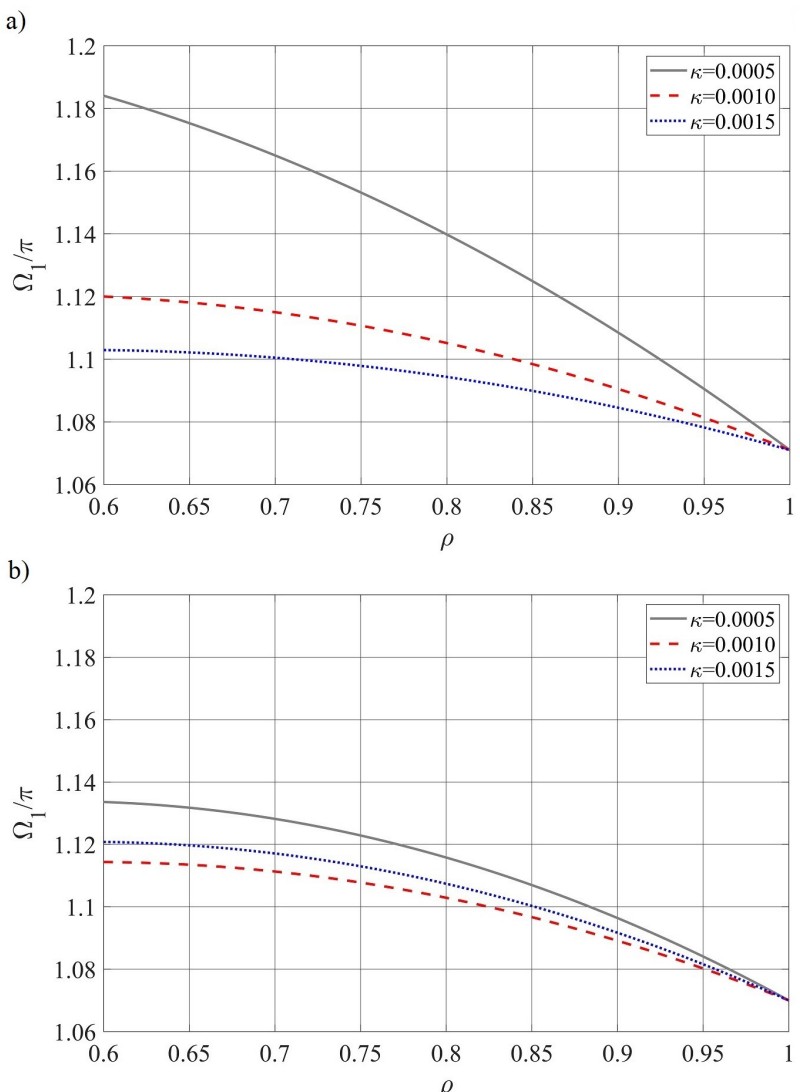

b)

**Figure 12.** Effect of the length ratio on the non-dimensional fundamental frequency of a two-cable network ($\varepsilon = 0.5$): (**a**) strain-softening cross-tie behavior; (**b**) strain-hardening cross-tie behavior.

## 5. Conclusions

A cross-tie is an effective solution to suppress unfavorable motions of stay cables on cable-stayed bridges. One major drawback of this measure is its inability to directly dissipate energy from the formed cable network. To address this limitation, employing materials with high energy-dissipation features for cross-ties has been proposed, of which the presence of material nonlinearity cannot be ignored. Although the nonlinearity in the cross-tie restoring force resulting from slackening and snapping has been investigated in a few studies, research on the effect of cross-tie material nonlinearity on the dynamic response of the formed cable network is scarce. In this study, the authors have expanded upon an existing analytical model of a two-shallow-flexible-cable network to incorporate the nonlinear behavior of the cross-tie material. The cross-tie material properties are captured using a piecewise power series polynomial. The application of the Harmonic Balance Method (HBM) facilitates the determination of the equivalent linear stiffness of the cross-tie. The dynamic response of a cable network with a nonlinear cross-tie is approximated by evaluating that of an equivalent linear system. The modal responses of cable networks yielded from the proposed analytical model and the HBM-based approach manifest as good correspondence with those obtained based on an FE simulation. In addition, the effects of the cable vibration amplitude, cross-tie material property, and

installation location, as well as the length ratio between the neighboring cable and the target cable, on the cross-tie equivalent linear stiffness and the fundamental frequency of the cable network have been investigated and discussed. It is worth mentioning that in the proposed method, the dynamic behavior of a nonlinear cable network system is approximated by an equivalent linear system, of which the vibration amplitudes of main cables are assumed to be insignificant when compared to the cable length. Thus, the outcome of the current study may not be representative of a nonlinear cable network at a very large vibration amplitude. The main findings are concluded as follows:

1. For a cable network with a general layout, the presence of cross-tie material nonlinearity would not only affect its modal frequencies and order, but also considerably alter the mode shapes. When the cross-tie material is of the strain-softening type, both the cable network fundamental frequency and the cross-tie equivalent linear axial stiffness show a monotonic decrease pattern with the increase in cross-tie axial deformation. On the other hand, if a strain-hardening type of material is used for the cross-tie, these two parameters are observed to exhibit a decrease–recover pattern.

2. Compared to the linear cross-tie case, when a strain-softening type of cross-tie is used in a cable network with a general layout, the modal frequencies of all the modes would decrease. The modal order of the lowest in-phase global mode is not affected and the associated mode shapes remain more or less the same. However, some high-order global modes would evolve into local modes dominated by one of the main cables, and out-of-phase global modes are excited. In addition, the formation of local modes is delayed to a high order. A local segment mode could evolve into a global mode or a local mode dominated by one of the main cables.

3. In a twin-cable network, the presence of cross-tie material nonlinearity has no impact on the modal frequencies and modal order of all symmetric in-phase global modes. However, when a strain-softening type of cross-tie material is used, the anti-symmetric in-phase global modes delay to a higher order with the same modal frequencies. In addition, all local segment modes evolve into anti-symmetric out-of-phase global modes with reduced modal frequencies.

4. In contrast to the linear cross-tie case, when a cross-tie exhibits nonlinear behavior, moving it towards the cable mid-span was found to be disadvantageous to the enhancement of the cable network in-plane stiffness. This occurs regardless of whether a strain-softening or a strain-hardening type of material is used for the cross-tie.

5. A cable network using a strain-softening or strain-hardening type of material for the cross-tie would respond differently to the change in the cross-tie location. The fundamental frequency and cross-tie equivalent linear axial stiffness in the former case are less sensitive to the cross-tie position effect than that in the latter.

6. The increase of the length ratio would lead to less axial deformation in a cross-tie when the cable network vibrates in its fundamental mode. Within the length ratio range covered in the current study, the presence of a threshold length ratio has been identified in the case of a strain-softening-type cross-tie, where the equivalent linear stiffness of the cross-tie is observed to increase drastically beyond this threshold. In addition, the effect of the length ratio on the network in-plane stiffness and frequency is determined by the level of its influences on the cross-tie axial stiffness and the neighboring cable lateral stiffness. A change in the cross-tie equivalent linear stiffness would not necessarily result in a change in the actual stiffness of the entire cable network system with the same trend.

**Author Contributions:** A.Y.: Conceptualization, Methodology, Formal analysis, Investigation, Software, Validation, Data curation, Visualization, Writing—original draft. S.C.: Supervision, Conceptualization, Methodology, Writing—review & editing, Funding acquisition. All authors have read and agreed to the published version of the manuscript.

**Funding:** The financial support provided by the Natural Sciences and Engineering Research Council of Canada (NSERC) (RGPIN-2022-02973) for this project is greatly appreciated.

**Data Availability Statement:** Not applicable.

**Conflicts of Interest:** The authors declare that they have no known competing financial interest or personal relationships that could have appeared to influence the work reported in this paper.

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
