# Peer review of "Impact of Cross-Tie Material Nonlinearity on the Dynamic Behavior of Shallow Flexible Cable Networks"

_computation, doi:10.3390/computation11090169_

Round 1
Reviewer 1 Report
This work presents new extension of alternative analytical model of a two shallow-flexible-cable network, differing by ability of taking into account the cross-tie material nonlinearity. Authors performed its verification by rigorous finite element method via commercially available software. By using proposed approach, they successfully simulated and researched responses of cable network under linear and non-linear cross-ties under various operation modes. The work corresponds to aims and topics of the Special Issue of “Computation”. It is suitable for publication in the present form.
Author Response
We are sincerely grateful for your time and insightful feedback on my paper. Your positive comments and acceptance of the present format are both encouraging and motivating.
Reviewer 2 Report
The article deals with an important problem related to the damping of vibrations of the cables supporting the bridge. In particular, it is related to the determination of impact cross-tie material non-linearity on the dynamic behaviour of shallow flexible cable networks. While many analyses have been done to date on the dynamics of cable networks, the impact of non-linearity of the cross-tie material has not yet been assessed. In the conducted analysis, the authors proposed a new analytical model of two-shallow-flexible-cable network taking into account the nonlinearity of the cross-tie material. The dynamic response of a cable network containing a nonlinear cross-tie was approximated by an equivalent linear system obtained using the harmonic balance method (HBM). Two different types of material non-linearity were considered, including strain softening and strain hardening behaviour. To verify the validity of the proposed analytical model, finite element simulations using Abaqus software were conducted to compare the numerical results with those obtained from the proposed analytical model. The modal frequencies of the twin-cable network predicted by the finite element simulation and those obtained by the proposed analytical model demonstrate good agreement.
In addition, the authors conducted parametric studies and investigated the effect of cable vibration amplitude, cross-tie material property, and installation location, as well as length ratio between the consisting cables on the cable network fundamental frequency and the cross-tie equivalent linear stiffness.
In the article, the authors showed significant discrepancies between the dynamic response of the cable network with the nonlinear cross-tie, and on the linear cross-tie assumption. The results obtained suggest that neglecting the non-linear behaviour of the cross-tie can result in a misleading prediction of the dynamic response of the cable network.
In conclusion, the presented paper is original. The analysis procedures are very clear and comply with the applicable methods. The conclusions presented are consistent with the evidence and arguments presented in the paper. Finally, I recommend this article for publication.
Minor editing of English language required
Author Response
Your recommendation for publication is truly appreciated and serves as a testament to the quality of the research. Thank you for your time, and support. We want to assure you that all English-related issues will be carefully checked to enhance the clarity of the paper.
Reviewer 3 Report
General Comment
The manuscript presents an extensive analytical study on the effect of the material nonlinearity of cross-ties in the dynamic behavior of cable networks, as the ones used in stayed bridges. For this, an existing analytical model for a network incorporating two cables is extended in order to include the nonlinear force-displacement relationship of the cross-tie. The harmonic balance method is used to derive an equation to compute an equivalent linear stiffness for the cross-tie, in order to approximate the cable network to an equivalent linear system. The analytical model is validated against the dynamic behavior of two cables networks determined from nonlinear models using finite element analysis. Then, a parametric study is performed in order to study the effect of the cross-tie nonlinearity (with strain softening and strain hardening), the cross-tie position, and the cable length ratio, in the dynamic behavior of a two cables network. The results are presented and discussed. Among the several results, it is concluded that the material nonlinearity of cross-ties can significantly impact the dynamic behavior of cable networks.
The topic of the manuscript is very interesting. The dynamic behavior of cable networks in stayed bridges due to wind, and other effects, can be critical for design, namely for the safety and service life. The influence of the cross-tie nonlinearity in such cable systems still need further studies and this manuscript constitute a good contribution for the field. The presented results could be useful for future studies and also as a guide for designers of stayed bridges.
I made some comments/suggestions in order to improve the manuscript. The authors should take the comments into account and revise their manuscript.
Specific Comment 1
The affiliation of the authors is missing, as well as the required information in the end of the manuscript and before the references (Author Contributions, Funding, …). Please refer to the mdpi guidelines for authors
Specific Comment 2
In the Keywords, “dynamic behavior” should also be added.
Specific Comment 3
For the sake of the readers, in the Introduction section please add a picture or photo of a cross-tie connecting stay cables.
Specific Comment 4
In Section 3.1, please add a picture of one of the used FE models (with undeformed and deformed shape).
Specific Comment 5
In Section 3.1, please explain how the effect of the cable pretension was simulated in the FE model. Was this effect simulated with a negative and uniform temperature variation?
Specific Comment 6
The graphical quality and readability of several figures need to be improved.
Specific Comment 7
The Conclusion section is too long and should be shortened. I suggest that, after an introductory paragraph, the main conclusions should be summarized in a point-by-point format.
Specific Comment 8
In the end of the Conclusion section, and based on your results, please add a small paragraph with some guidelines for design.
